# Effects of Stocking Density on Growth Performance and Stress Responses of Bester and Bester ♀ × Beluga ♂ Juveniles in Recirculating Aquaculture Systems

**DOI:** 10.3390/ani11082292

**Published:** 2021-08-03

**Authors:** Lorena Dediu, Angelica Docan, Mirela Crețu, Iulia Grecu, Alina Mogodan, Marilena Maereanu, Lucian Oprea

**Affiliations:** 1Faculty of Food Science and Engineering, “Dunărea de Jos” University of Galați, 800008 Galați, Romania; angelica.docan@ugal.ro (A.D.); iulia.grecu@ugal.ro (I.G.); alina.antache@ugal.ro (A.M.); 2Romanian Center for Modelling Recirculating Aquaculture Systems, “Dunărea de Jos” University of Galați, 800008 Galați, Romania; lucian.oprea@ugal.ro; 3S.C. Danube Research-Consulting S.R.L., 825200 Isaccea, Romania; marilena.maereanu@gmail.com; 4Cross-Border Faculty, “Dunărea de Jos” University of Galați, 800008 Galați, Romania

**Keywords:** stocking density, hybrid sturgeons, overcrowding stress, physiological response, RAS, serum parameters

## Abstract

**Simple Summary:**

The disappearance of many wild sturgeon populations from natural waters and the high demand for sturgeon products (meat and caviar) have led to increased aquacultural production of sturgeons. The development of recirculating aquaculture systems requires the optimization of rearing technologies so that high stocking densities have a minimal impact on fish health. Due to the ease with which sturgeons hybridize, the optimization of sturgeon aquaculture technologies involves the identification of hybrid lines with high technological performance and high stress resistance. One of the best known hybrids is that between the beluga and the sterlet (bester). The present study aims to evaluate the hybrid obtained by crossing the bester (female) with the beluga (male) as a new candidate for the recirculating aquaculture systems. The results show better performance and better density adaptation of the bester × beluga hybrid compared with the bester hybrid. However, at higher stocking densities, a negative impact on growth performance and physiological response was observed.

**Abstract:**

The study aimed to compare the growth performance and physiological responses of bester (B) and backcrossed bester ♀ × beluga ♂ (BB) in response to crowding stress under different stocking densities, as well as to establish a threshold stocking density for rearing BB in a recirculating aquaculture system (RAS) without welfare impairment. For this purpose, in the first trial (T1), B (181.15 ± 21.21 g) and BB fingerlings (181.98 ± 28.65 g) were reared in two stocking densities of 2 kg/m^2^ and 4 kg/m^2^ in fiberglass tanks (1 m^3^) for 6 weeks. In a parallel trial (T2), the BB hybrids (335.24 ± 39.30 g) were kept in four initial stocking densities, ranging from 5 kg/m^2^ to 12 kg/m^2^. The results of T1 revealed better growth indices (i.e., final mean weight, weight gain, specific growth rate) at lower stocking densities for both hybrids; however, in terms of growth performance, the BB hybrid showed better results when compared with the B hybrid. BB hybrids registered significantly (*p* < 0.05) lower serum cortisol and MDA and higher lysozyme than B hybrids, showing higher tolerance to crowding stress. Nevertheless, at higher densities, selected serum parameters (i.e., hematological indices, cortisol, glucose, protein, malondialdehyde, lysozyme) and growth performance indices used to evaluate the hybrids indicate that high stocking density could affect the growth and welfare of BB hybrids, and that the selected serum parameters could be used as good indicators for chronic stress caused by overcrowding conditions.

## 1. Introduction

The fast development of sturgeon farming is closely associated with progress in technology, resource conservation, acclimatization, artificial propagation, culture practices, and management [1]. In the actual context of limited access to land resources and suitable water supply, climate change, and increased demand for diversification, recirculating aquaculture systems (RASs) emerged as promising, versatile, and highly productive systems. In RASs, through control of temperature, light, feeding regime, and water quality, sturgeons reach marketable sizes faster [2,3] and attain maturity very quickly—up to twice as fast as in their natural environment [4]. Nevertheless, RASs are a costly engineering approach, with high initial investment to install and later operate [5]. It is favorable for farmers to increase stocking density, as this is a key factor in aquaculture that may enable them to increase their final yield. However, fish welfare and performance may be affected by chronic stress under crowded conditions [6]. Therefore, sturgeon farmers practicing caviar production are looking for genetically improved lines to enhance not only technological performance, but also tolerance for high densities. This last feature has greater importance, especially in species maintained for extended periods—such as sturgeons—where chronic stress is associated with viral, bacterial, and fungal disease outbreaks [7], as well as changes in behavior, maturity [8], and ovarian cycles [9].

Although initially sturgeon aquaculture was dedicated mainly to caviar production, in recent decades, demand for sturgeon meat has shown a positive trend with consumer acceptance [10], and this trend will be accentuated as the level of awareness of its undeniable nutritional values increases [11]. In caviar farms, females can be identified after 2–5 years when, depending on the species, they begin to differentiate sexually [12]. Therefore, for some larger species with late maturity, the males are commercialized long after they reach market size, which causes a decrease in profit of 30–40% [13]. In this context, sturgeon hybrids are becoming more and more popular for the aquaculture industry due to better growth performance compared to parental species, or to other traits, such as earlier maturation, both of which represent a premise for their potential in meat and caviar production.

Hybrid bester (B)—a cross between beluga (*Huso huso* Linnaeus, 1758) and sterlet (*Acipenser ruthenus* Linnaeus, 1758)—is characterized by an excellent growth rate under aquaculture conditions, which may even surpass the rapid growth of the maternal species [14]. Bester produces high-quality caviar at a younger age compared to beluga [15]—a trait which places bester in the shortlist of the most popular hybrid sturgeons. Backcross hybridization of bester with the parental species has been previously practiced in Russia [16], but there is limited scientific information on the subject. Beluga is a fast-growing sturgeon tolerant to high stocking densities and unfavorable rearing conditions [14]; however, a long maturation period is a disadvantage [17]. Backcrossing of F1 bester females with Danube beluga males (BB hybrids) was performed in a Romanian farm, with the purpose of obtaining hybrids with increased growth rates and higher tolerance for crowded conditions. If these traits were demonstrated, the BB hybrids would allow the shortening of the meat production cycle in sturgeon farms, providing financial sustainability for companies enrolled in caviar production, until the first harvest. To the best of our knowledge, there is no evidence in the scientific literature regarding the rearing conditions or stocking density limits of the above-mentioned hybrid.

Stocking density is recognized as an important technical factor with a high impact on fish welfare and productivity. Many studies have reported the effects of stocking density on teleost fish [18,19,20,21,22], but few have addressed sturgeon species [23,24,25,26,27,28]. However, for Ponto-Caspian sturgeons and their hybrids, there is even less information available [29,30,31]. Stocking density may act as a biological stressor, causing the development of a physiological response for the maintenance of internal homeostasis [32,33]. This adaptive response is expressed as an interrelated cascade of signals, affecting the organism’s functions at primary levels: determined by the endocrine changes mostly regarding the circulating catecholamines and stress hormones (corticosteroids) that directly affect the next physiological level; secondary levels: regarding the changes in glucose, total proteins, antioxidant capacity, hydromineral balance, hematological indices, the immune system (lysozymes), etc.; and tertiary levels: related to the changes in growth performance, disease resistance, and behavior that can affect the whole-animal integrity and survival [34]. Taking all of these into account, the use of blood analysis for monitoring the physiological status of fish under stressful conditions has been suggested as a minimally invasive method, especially when working with economically valuable sturgeons [35]. In general, sturgeon responses to stressors are not as high as in teleost fish [36], but these differences—even between closely related species—indicate that there are species-specific variations regarding crowding tolerance among this group of primitive fish [37]. Other studies have reported adverse effects of high stocking density on different sturgeon species, such as beluga sturgeon [25], Atlantic sturgeon (*Acipenser oxyrinchus* Mitchill, 1815) [26,38], and Amur sturgeon (*Acipenser schrenckii* Brandt, 1869) [39,40].

Enhanced growth rate and stress resistance are the most desirable traits for stock improvement in aquaculture. The hypothesis of the study was that the hybrids obtained by backcrossing bester hybrids (F1) with purebred belugas are eligible for RAS production, having higher growth performance than bester—a trait necessary for the financial sustainability of a sturgeon farm.

Thus, this study aimed to assess the potential of bester x beluga hybrids for intensive production under RAS conditions. The secondary objectives of this study were to (1) compare the growth performance of bester (B) and backcrossed hybrids of bester females × beluga males (BB) under different stocking densities; (2) assess the primary, secondary, and tertiary responses to crowding stress in both hybrids; and (3) evaluate the effect of stocking density up to 12 kg/m^2^ in relation to growth, hematological, and biochemical parameters of the BB hybrids.

## 2. Materials and Methods

### 2.1. Experimental Design and Fish Maintenance

The present study was conducted in a pilot recirculating aquaculture system (RAS) equipped with automatic water treatment facilities (Appendix A), located at the Romanian Center for Modelling Recirculating Aquaculture Systems from “Dunărea de Jos” University of Galați. The conceptual installation framework of the RAS was designed to ensure proper technological treatment for water quality and fish biomass welfare. The flux scheme includes a well-thought-out sequence of components displayed on two levels (basement, and ground floor). There are main components (1), including growth units (24 tanks, each of 1 m^3^, 109 cm in diameter), mechanical filters, biological filters, equipment for transfer of dissolved gases (degassing carbon dioxide, oxygen injection, and contactors), disinfection systems with UV radiation, pumps, and water quality monitoring and control equipment; as well as auxiliary components (2), such as automatic feeders, an ozone generator, and an independent electric generator.

The hybrid sturgeons were obtained at the hatchery of a commercial farm belonging to Danube Research Consulting (DRC), via artificial reproduction using a stock of 9 aquaculture F1 breeders (4 bester females, and 5 males, of which were 3 besters and 2 belugas). Before fertilization, the eggs from each female were divided into two groups, each group being fertilized with the pooled milt of males of the same hybrid/species. The obtained larvae were reared in a flow-through system until they reached the age of 3 months. Prior to the experiments, the fish were reared in a pilot RAS station until the age of 5 months, when they were used for experiments. The fish were sorted to narrow the size within each group, and to allow for small variations at the start of the experiment. The experiment was set as two parallel trials, called T1 and T2. For the first trial (T1), 90 B hybrids with an average initial weight of 181.15 ± 21.21 g and 90 BB hybrids with an average initial weight of 181.98 ± 28.65 g were each randomly distributed into 6 fiberglass tanks (1 m^3^) in 2 different densities (2 kg/m^2^ (LD) and 4 kg/m^2^ (HD)) for 6 weeks (3 replicates for each hybrid; 12 tanks in total). For the second trial (T2), 196 BB hybrids (335.24 ± 39.30 g/specimen) were distributed in 8 tanks (1 m^3^) at 4 stocking densities (in duplicate): 5 kg/m^2^—SD1; 8 kg/m^2^—SD2; 10 kg/m^2^—SD3; and 12 kg/m^2^—SD4. The second trial was performed in parallel with the first trial.

Fish were fed three times daily (2% of tank biomass per day) with commercial pellets for sturgeons (54% protein, 15% fat, 0.5% fiber, ash, and 21.1 kJ·g^−1^; Alltech Coppens, Leende, the Netherlands). The feed was administered manually to avoid feed competition and fighting amongst fish. When uneaten pellets were observed, the feed administration was stopped, and the remained pellets were quantified and dried after removal from the tank using a small net. Body weight and standard length were recorded weekly for 10 fish/tank (chosen at random), and the feeding rates were adjusted to the new biomass of each rearing unit.

### 2.2. Water Quality Parameters

Water quality parameters were monitored daily during the trials. Water temperature, pH, dissolved oxygen, ammonium, and nitrate were measured automatically with an Endress+Hauser monitoring system (Endress + Hauser AG, Reinach, Switzerland) (probes of oxygen and temperature were placed in each tank while the RAS system was provided with two probes for pH, ammonium, and nitrate) and double-checked using a pH meter (HI 81143, Hanna, Cluj-Napoca, Romania) and oxygen meters (HI 9142, Hanna). Nitrogen compounds were quantified weekly with a Skalar SAN++ analyzer (Skalar Analytical, the Netherlands) according to the manufacturer’s instructions. The physical and chemical characteristics of the water were similar between tanks for both trials. For the first trial, the mean temperature was 20.23 ± 0.45 °C, pH was 7.45 ± 0.14, total ammoniacal nitrogen 0.28 ± 0.13 mg/L, un-ionized ammonia was 0.018 ± 0.009 mg/L, and dissolved oxygen was 7.54 ± 1.03 mg/L. For the second trial, the mean temperature was 20.71 ± 0.45 °C, pH was 7.39 ± 0.20, total ammonia nitrogen was 0.19 ± 0.23 mg/L, un-ionized ammonia was 0.02 ± 0.01 mg/L, and dissolved oxygen was 7.36 ± 1.13 mg/L. All parameters were considered adequate for sturgeon aquaculture [12]. The photoperiod was 10:14 h light:dark. The light source was naturally enhanced with fluorescent light (equally distributed above the rearing units), providing a light intensity of 160 lx during the daylight hours.

### 2.3. Growth Performance

At the end of each trial, fish were subjected to fasting for 24 h, and all individuals were assessed for their body weight (BW) and standard length (SL). Growth performance and feed utilization parameters were calculated according to the following equations:Weight gain (WG, %) = [(BWf − BWi)/BWi] × 100, where BWi and BWf are the initial and final average body weight (g) of fish sampled from each tank, and t is the experimental period in each trial (day);Specific growth rate (SGR, %/day) = [(lnBWf − lnBWi)/t] × 100;Fulton’s condition factor (K, %) = [BWf (g)/(SL (cm)^3^] × 100, where SL = standard body length;Feed conversion ratio (FCR) = FI (g)/BG (g), where FI stands for food consumption (food provided – uneaten food) and BG is biomass gain per tank;Protein efficiency ratio (PER) = BG/protein consumed;Hepatosomatic index (HSI) = [liver weight (g)/body weight (g)] × 100);Viscerosomatic index (VSI) = [visceral weight (g)/body weight (g)] × 100);The coefficient of variability (CV) = CvBW (%) = 100 (SD/mean BW), and was calculated for the body weight on the initial (CvBWi) and final (CvBWf) days of the experiment.

### 2.4. Sampling Protocol and Blood Analysis

At the end of the experimental period, 10 fish were randomly sampled from each tank to evaluate hematological variables. In order to reduce handling stress, the fish were anesthetized with 2-phenoxyethanol (0.7 mL/L) until deep anesthesia. The anesthetic was selected based on the criterion of having no effects on the hematological profile [41]. Fish were quickly captured and the blood samples were taken from the caudal vein using a heparinized syringe and transferred to sterilized tubes. The procedure was performed on ice until samples were transferred to the laboratory for further analysis. For each tank, five fish were dissected for liver and visceral weight measurements.

The hematological profile was determined using the routine methodology of fish hematology. The red blood cell counts (RBC × 10^6^/μL) were determined with a Neubauer hemocytometer using a Potain pipette and Vulpian diluting solution (prepared in house from sodium citrate, potassium iodide and metallic iodine (Sigma-Aldrich, St. Louis, MO, USA)). The red blood cells were counted from 5 small squares of the hemocytometer [42] using a Zeiss Axio Imager research microscope (Zeiss International, Thornwood, NY, USA). The hemoglobin concentration (Hb, g/dL) was measured by a colorimetric method using Drabkin’s reagent (DIALAB, Wiener Neudorf, Austria), and then the absorbance was read at a wavelength of 540 nm [43] using a Specord 210 UV–Vis spectrophotometer (Analytic Jena, Jena, Germany). To determine the hematocrit (PVC %), capillary tubes and a Hettich Haematokrit 210 centrifuge (Hettich Zentrifugen, Tuttlingen, Germany) were used. The blood was centrifuged for 5 min at 12,000 rpm (13,709× *g*) [43]. The hematological indices, mean corpuscular volume (MCV, fL), mean corpuscular hemoglobin (MCH, pg), and mean corpuscular hemoglobin concentration (MCHC, g/dL) were determined as described in the literature [44]. For biochemical assays, blood samples were immediately centrifuged for 10 min at 3500 rpm (1166× *g*) in a Hettich Mikro 120 (Hettich Zentrifugen, Tuttlingen, Germany), and then plasma was separated in 1.5-mL Eppendorf tubes. The VetTest^®^ Chemistry Analyzer, using IDEXX VetTest kits (IDEXX Laboratories, Inc., Westbrook, ME, USA), was used to determine glucose concentration (GLU mg/dL) and total protein (TP g/dL) in plasma. Lipid peroxidation (malondialdehyde-MDA nmol/mL) was performed in accordance with the Draper and Hadley method [45], at an optical density of 532 nm (SPECORD 210). Serum lysozyme activity was measured based on the turbidimetric assay, Enzymatic Activity of Lysozyme Protocol (Sigma, EC 3.2.1.17, Sigma-Aldrich, St. Louis, MO, USA). In brief, 66 mM of potassium phosphate buffer (pH 6.24 at 25 °C) was mixed with a volume of 0.01% (*w*/*v*) suspension of *Micrococcus lysodeikticus* (Sigma, M3770, Sigma-Aldrich, St. Louis, MO, USA). Lyophilized powder of chicken egg white lysozyme (Sigma, L6876, Sigma-Aldrich, St. Louis, MO, USA) was used as a standard. One unit of lysozyme activity was defined as a reduction in the absorbance of 0.001/min, at a 450-nm wavelength. Serum cortisol determination was performed using the Fish Cortisol kit (CUSABIO Biotech Co, Houston, Texas, USA), based on a competitive immunoenzymatic colorimetric method for the quantitative determination of cortisol in fish serum. Absorbance was read at 450 nm using an ELISA microplate reader (Tecan Sunrise, Tecan GmbH, Grödig, Austria).

### 2.5. Data Analysis

The data were analyzed using SPSS for Windows, Version 16.0 (SPSS Inc., Chicago, United States). Growth indices and hematological and serum parameters were expressed as means ± SEM of the replicates, considering each tank as an experimental unit. In the first trial, data were analyzed by two-way ANOVA analysis, with stocking density and hybrids as independent variables. Before statistical analyses, both normality and homogeneity of variance were confirmed by Shapiro–Wilk and Levene’s tests, respectively.

For the second trial, to describe the relationships between stocking density and different variables, the data were subjected to regression analysis to fit the best model based on *p*-value. The significance of regressions was tested using ANOVA. The level of significance was set at *p* < 0.05 for all analyses.

## 3. Results

### 3.1. Trial 1 (T1)

#### 3.1.1. Growth Performance

Zootechnical performance and feed utilization parameters of hybrids from T1 are presented in Figure 1 and Table 1. There were no mortalities during the trial.

The final weight of BB hybrids was significant higher (*p* < 0.05) than B hybrids in both density groups (Figure 1). Better values of FCR and PER were observed for LD groups in both hybrids. WG and SGR showed no significant differences between LD and HD groups for BB backcrosses, and significant differences for B hybrids. However, BB backcrosses showed better results in both density groups when compared with B hybrids (Table 1).

Regardless of stocking density, values of K, VSI, and HSI were significantly different between the two hybrids, while only for besters were HSI and VSI values significantly different between densities (*p* < 0.05).

#### 3.1.2. Hematological Parameters and Blood Indices

The hematological parameters of B and BB backcrosses reared at two different stocking densities are presented in Table 2. All measured indices, except for PCV (for B hybrids) and Hb level (for BB hybrids), were significantly different (*p* < 0.05) between densities, in both hybrids (Table 2). A significant increase in RBCs was observed in HD for both hybrids; BB showed the highest value in the HD group. Stocking density significantly decreased Hb values for B hybrids (*p* < 0.05), while in BB hybrids no significant differences were detected (*p* > 0.05). The PCV values decreased for both hybrids, but were significantly lower (*p* < 0.05) only in BB hybrids reared at HD. All hematimetric indices showed significantly lower values in HD in both hybrids, except for MCHC values registered for BB hybrids, which showed significantly lower values in LD groups.

The cortisol level was slightly higher in crowded groups; however, no significant differences (*p* > 0.05) were found between densities for both hybrids. The level of cortisol in B hybrids was significantly higher than in BB hybrids (Figure 2).

Although the level of serum glucose was slightly higher for both hybrids held in HD groups, there were no significant statistical differences between densities (*p* > 0.05). Nevertheless, in both density groups, serum glucose was significantly higher in B hybrids than in BB hybrids (Figure 2). Lysozyme and total protein levels registered no significant difference (*p* > 0.05) in HD groups compared with LD groups; there were, however, significant differences (*p* < 0.05) between the two hybrids when the same density groups were compared. MDA was slightly increased in HD groups for both hybrids, but only for the B hybrid was the difference between HD and LD statistically significant (*p* < 0.05) (Figure 3).

### 3.2. Trial 2 (T2)

#### 3.2.1. Growth Performance

Weight gain and growth parameters of BB backcrosses reared at different densities are presented in Table 3. There were no mortalities during the trial (except for one fish in SD1, due to an accident).

A quadratic effect induced by density was found for FW, WG, and SGR, with the best values in fish held at the lowest density. Linear regression analysis was the best model describing relationships between the stocking density and feeding efficiency indicators PER and FCR. Thus, a positive linear relationship was found for FCR and stocking density, while PER linearly decreased with density.

Increasing stocking density also had a quadratic effect on HSI and VSI, while final length and K did not show a significant linear or nonlinear regression. Final body weight variability, measured at the end of T2, increased with an increase in density.

#### 3.2.2. Haematological Parameters and Blood Indices

The hematological indices and serum biochemical parameters measured in BB hybrids after 6 weeks of rearing in densities up to 12 kg/m^2^ are presented in Table 4.

Increasing stocking density caused a proportional linear increase in RBC count, while quadratic regression was applied for the other hematological parameters. However, although Hb and hematimetric indices (MCH and MCHC) showed a relatively strong quadratic relationship with stocking density, those models were not statistically significant (*p* > 0.05).

Protein concentration varied between the highest for SD1 (4.57 g/L) and the lowest for SD3 (3.88 g/L), and for this parameter quadratic regression was the best fit for the data. Lysozyme activity showed a negative linear regression, decreasing from 10.50 ± 1.61 U/mL (SD1) to 8.78 ± 0.62 U/mL (SD4). In contrast, a clear increase in serum malondialdehyde formation in fish was observed in the SD4 group. Quadratic regression was is the model that best described the relationship between density and MDA. Glucose and cortisol increased slightly with increasing stocking density, but no significant relationship was found for any of the variables.

## 4. Discussion

For the aquaculture industry, stocking density is a factor of great importance, affecting fish welfare and causing health problems when applied inappropriately for the cultured species. Until now, few studies have evaluated the impact of stocking density on bester hybrids, while there are no studies that we are aware of regarding the effects of stocking density on the bester × beluga hybrid cultured in an RAS.

The results of this study show that both kinds of hybrid sturgeon perform better under lower stocking densities in RASs. Retardation in growth due to high density stress has been reported in different sturgeon species [23,25,26,39,46,47]. This has already been demonstrated for some teleost species—such as European seabass (*Dicentrarchus labrax* Linnaeus, 1758) [48], rainbow trout (*Oncorhynchus mykiss* Walbaum, 1792) [49], Nile tilapia (*Oreochromis niloticus* Linnaeus, 1758) [50], turbot (*Psetta maxima* Linnaeus, 1758) [51], and Atlantic cod (*Gadus morhua* Linnaeus, 1758) [52]—for which high stocking densities reduced growth. In many cases, though, it is hard to separate the direct negative effects of stocking density from the secondary causes associated with density, such as water quality deterioration as a result of ammonia and nitrite accumulation, oxygen depletion, social interaction, and lower feed intake [53]. Under the present experimental conditions, water quality was not a limiting factor, since the RAS system used for the experiments is fully automated, assuring oxygen supply if a decrease in O_2_ concentration was detected by optical probes. Feed was introduced manually to minimize competition between individuals, and water quality parameters were monitored and exchanged with fresh dechlorinated water as needed.

Our results showed higher values of FCR in HD groups, indicating that fish were not able to utilize the given feed due to the chronic stress induced by stocking density. Similar results were reported for Atlantic sturgeon by Jodun et al. [26] after 7 weeks of feeding fish with a ratio of 2.5% BW (initial weight of 368.7 g/specimen, reared in densities ranging from 3.6 to 10.9 Kg/m^2^). In line with our results, Rafatnezhad et al. [25] showed the adverse effect of higher stocking density on beluga growth and feed utilization (initial weight of 93.13 ± 1.04 g/specimen reared in densities ranging from 1 to 8 kg/m^2^). The growth of sturgeons may be also suppressed at high stocking densities [26,54], although in these cases, some hematological parameters were less affected compared with teleost fish. In contrast with the above-mentioned studies, it has been demonstrated that for some early-stage sturgeons (larvae and fingerlings)—such as lake sturgeon *(Acipenser fulvescens* Rafinesque, 1817) and Atlantic sturgeon (*Acipenser oxyrinchus*)—growth performance did not exhibit significant differences at different densities [55,56]. Nevertheless, for juvenile Siberian sturgeons (9.20 ± 0.34 g) (*Acipenser baerii* Brandt, 1869), higher densities (up to 16.60 kg/m^2^) enhanced growth performance [57].

The results from the T1 trial reveal that, despite being genetically similar, there are hybrid-specific differences regarding the effect of stocking density on growth performance between B and BB backcross sturgeons. In HD groups of B hybrids, significant effects of density were observed on WG, SGR, PER, and FCR, while for BB backcross sturgeons only FCR and PER were negatively affected by density. Better feed conversion at the lowest density may be related to higher swimming activity observed in the crowded groups. However, comparing the growth indices of the two hybrids in T1, we can state that BB backcross sturgeons showed better performance than B hybrids in both density groups.

The results of T2 showed that increasing density proportionally increased FCR (from 0.91 to 1.15) and decreased SGR (from 1.59 to 1.26). Previous studies have also demonstrated that high stocking density can affect fish growth performance even when water quality in aquaculture systems is well maintained [58,59].

Social hierarchy, leading to an increased number of interactions, may have an important impact on growth performance, especially in groups with high variability. Adult sturgeons rarely manifest cannibalism, but in beluga juveniles, increased incidence was reported as stocking density grew [25]. At the end of both trials, higher weight variability was observed in higher density groups—particularly in BB hybrids where interactions were also present.

Some authors demonstrated that beluga sturgeon juveniles (from 93.13 g to 211 g) could be produced in densities of up to 8 kg/m^3^ without negative physiological consequences, but with lower growth performance [25]. Similarly, larger beluga (140–500 g) could be reared in densities up to 6 kg/m^3^, respectively up to 12 kg/m^3^ for fish weighing from 1500 to 3500 g with no negative impacts on somatic growth and feed efficiency parameters [37]. Other studies demonstrated that B hybrids could be reared successfully at densities of up to 15 kg/m^3^ with no significant effects on blood serum biochemical parameters [60]. For Siberian sturgeon—which is a temperature- and oxygen-tolerant species—the recommendations are that stocking density should not exceed 20 kg/m^2^ for juveniles up to 500 g [61]. Nevertheless, even though in some species stocking density may not induce growth alteration, changes at the physiological level should be detected early in order to avoid immune system suppression under overcrowded conditions.

Cortisol level is widely used in studying the effects of stress on fish, and is known as a standard stress indicator [62]. It has been suggested that cortisol is released into the circulatory system as a response to stimulation of the hypothalamic–pituitary–interrenal (HPI) axis by catecholamines, and is responsible for the mobilization of energy reserves by activating liver glycogenolysis and inhibiting glycolysis [63]. This, in return, results in a moderate increase in the plasma glucose concentration [64]. Cortisol and glucose levels are more sensitive to stress than the other variables of blood serum [65]. Although we noticed a significant difference between mean cortisol concentrations, for both densities, when the two hybrids were compared, a density of 4 kg/m^2^ was not a trigger for cortisol increase during T1. Similarly, in the second trial, no relationship was found between density and plasma cortisol levels. Similar studies showed that stocking density had no dramatic effect on plasma cortisol and glucose concentrations for beluga juveniles [30] and Siberian sturgeons [24]. In other studies, serum cortisol concentration in sturgeons held in high densities showed significantly higher values than those of lower density groups [13,33,66]. In sturgeons, unlike other fish species, cortisol levels are expected to be relatively low (2–20 ng/mL) [67,68]. However, large variations in cortisol values—between 0.4 and 60 ng/mL—are reported in the scientific literature for different sturgeons [24,69]. These values are highly dependent on species, the developmental stage of the fish, type or intensity of stress, and duration of stress. Similar cortisol values have been reported for juveniles of Chinese sturgeon (*Acipenser sinensis* Gray, 1835) (760.86 g) reared in an RAS system in densities up to 12.68 kg/m^2^ [33].

Glucose plays an important role in the bioenergetics of animals, since it is rapidly transformed to energy as ATP [70], and cortisol is one of the hormones mobilizing glucose production in fish through the gluconeogenesis and glycogenolysis pathways [71]. Although some authors consider glucose content to be a less precise indicator of stress [72], it has been widely used in chronic stress studies.

Increased levels of plasma glucose in fish under high stocking densities may reflect either the incapacity of these species to acclimate to crowded conditions [73], or the mechanisms behind acclimation [74]. In T1, density induced a slight increase in glucose, but no significant differences were found between densities. In T2, plasma glucose increased with density; however, it is important to underline that this increase was not statistically significant (*p* > 0.05). In contrast with our results, Long et al. [33] reported an increase in glucose concentration with stocking density. Other authors have reported lower glucose concentration values ranging between 41.4 and 46.8 mg/dL in beluga juveniles and 10 and 89 mg/dL in Siberian sturgeons kept at high stocking densities [24,25]. These studies, in line with our results, reported no significant effect of stocking density on blood glucose.

Hematological indices are often used in sturgeons as indicators of their overall physiological status, or as stress markers [75]. In T1, blood indices and hematological parameters showed a significant increase in RBC numbers in both hybrids held in HD. In BB backcross sturgeons, Hb was not affected by density, while decreased values were found for B hybrids. The reduction of hemoglobin can modify the oxygen quantity in tissues and, thus, lead to slower metabolic activity and lower energy production [76]. This mosaic of results from T1 shows that the adaptive response to stocking density is associated with the genetic background of the species. In T2, values of RBC and PVC increased with stocking density; the relationship of those parameters with density was characterized by linear and nonlinear (quadratic) regression, respectively. The threshold for MCV was situated between SD3 and SD4, decreasing from SD1 to SD3, while MCH and MCHC decreased in SD1 and SD2 but increased in SD3 and SD4. Similar results have been reported for belugas (143 ± 0.29 g) held under different stocking densities (1 to 6 kg/m^3^), where PVC was constant while RBC slightly increased after two months; however, these differences were not significant [29]. An accepted PCV range for fish is 20–45% [77], but sedentary bottom-dwelling fish have lower PVC than actively swimming species [52]. In our experiments, hematological values were within the ranges reported for other sturgeon species [25,30,78]. The elevations in Hb concentration, RBC count, and MCHC, accompanied by a diminished MCV, have been interpreted as adaptations to increased oxygen requirements [79] due to higher metabolic demands. The dynamics of the hematological parameters from the present study show the capacity of the hybrids subjected to a higher stocking densities to develop, under higher metabolic demand, an adaptive response to stress in order to maintain their hemostasis.

Lysozyme activity is widely used as an indicator of immune status in sturgeons [80]. In T2, lysozyme concentration significantly decreased in the sera of fish kept at higher densities, suggesting a weaker defense system in case of disease incidence. Similar results have been reported in Chinese sturgeons reared in an RAS, where lysozyme activity was decreased due to a high density of 12.68 kg/m^2^ [33].

Oxidative stress can be defined as a decline in the antioxidant defense mechanism. High stocking density can block the activity of metabolic and antioxidant enzymes, inducing significant oxidative stress and undesirable effects for fish welfare by disturbing the physiological equilibrium [81]. Malondialdehyde (MDA) is a lipid peroxidation metabolite, and is frequently used as a biomarker for assessing in vivo oxidative stress [82]. Some previous studies reported that high stocking density induces lipid peroxidation [20], while others have reported no significant difference in MDA levels at different stocking densities in largemouth bass (*Micropterus salmoides* Lacepède, 1802) [83], Senegalese sole (*Solea senegalensis* Kaup, 1858) [84], and tongue sole (*Cynoglossus semilaevis* Günther, 1873) [85]. In T1, MDA serum concentration measured in hybrids showed that a density of 4kg/m^2^ could induce oxidative stress in B hybrids. In T2, MDA measured in BB hybrids showed a quadratic response to density, decreasing from SD1 to SD2, and then increasing up to 91.12 nm/mL in SD4. Therefore, based on these results, a stocking density higher than 10 kg/m^2^ initiated oxidative stress in BB sturgeons.

In T1, total serum protein decreased in HD groups, although no significant differences were observed. Similar results have been reported for European seabass [86]. In T2, total protein fitted a quadratic regression model, showing a sharp decline from SD1 (5 kg/m^2^) to SD3 (10 kg/m^2^), followed by an increase in SD4 (12 kg/m^2^). A possible explanation for this is that proteins are rapidly utilized for metabolic purposes during periods of increased metabolic requirements. However, the recovery of total protein levels in higher densities, showing a possible adaptive response, does not exclude a change in protein fractions, with higher albumin and lower globulin content negatively impacting the fishes’ immunity [87].

Although both indices—HSI and VSI—only decreased in higher stocking densities for B hybrids during T1, the same tendency was also observed for BB hybrids in T2. This reduction seems to be associated with greater hepatic lipid utilization [10]. Moreover, the cortisol hypersecretion in stressed fish could change the blood distribution in internal organs [88]. Thus, the reduction in the blood volume from the strongly irrigated organs may cause a decrease in their volume, as was observed in our trials. In T2, the negative tendency of the morphological indices became more pronounced due to the higher densities—especially for SD4, where the highest level of cortisol was measured. Similar findings were reported for Chinese sturgeon in comparable stocking densities (3.7–9.0 kg/m^3^) [40]. In other species, high densities also reduced HSI and VSI [89,90], while Rafatnezhad et al. [25] reported no effects on the HSI of beluga sturgeons.

Final body weight variability, measured at the end of T2, showed a positive linear regression with density. This could be also related not only to density, but also to the genetic background of the hybrids.

## 5. Conclusions

Based on our findings, the BB backcross sturgeons showed a better performance than besters, suggesting that the heterosis effect was manifested in somatic growth. Therefore, in a short period of 6 weeks, the final weight of the BB hybrids was 13% higher than that of the besters. The present study showed that high densities could negatively impact the defense mechanisms (significant decreases in lysozyme activity), antioxidant status, and metabolism of BB hybrids. However, in terms of growth performance and feed utilization, BB backcross sturgeons could be regarded as a potential candidate for the aquaculture industry if appropriate technological management is applied. In order to mitigate the stress effects of stocking density, further research must be carried out to develop new strategies to enhance the immunological responses and adaptive mechanisms of fish to cope with crowding conditions.

## Figures and Tables

**Figure 1 animals-11-02292-f001:**
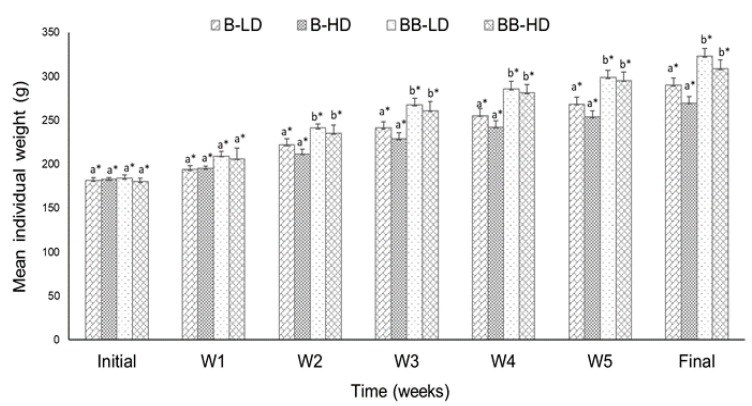
Growth dynamics for B and BB sturgeon hybrid juveniles reared at different densities during the 6-week study period. Values are presented as mean ± S.E.M (*n* = 10). Different letters indicate significant differences between hybrids from the same density group (*p* < 0.05; two-way ANOVA). */** Different symbols indicate significant differences between stocking density groups of the same hybrid.

**Figure 2 animals-11-02292-f002:**
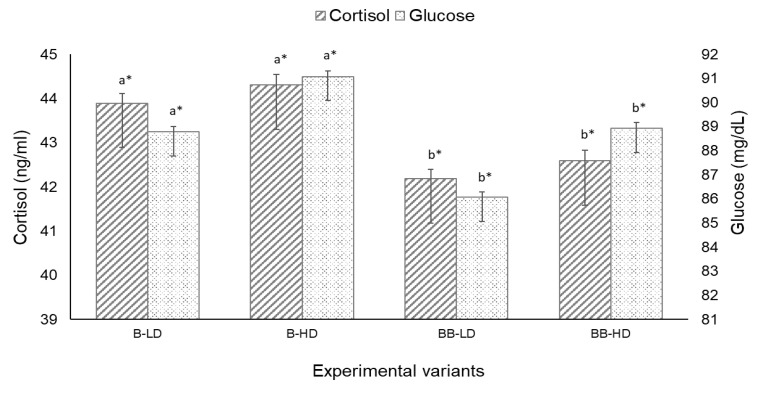
Serum cortisol hormone and glucose concentrations of bester (B) and bester × beluga (BB) sturgeon hybrids cultured in different initial stocking densities LD (low density—2 kg/m^2^) and HD (high density—4 kg/m^2^). Data are presented as mean ± SEM (*n* = 3 tanks per treatment). Different letters indicate significant differences between hybrids from the same density group (*p* < 0.05, two-way ANOVA). Different symbols */** indicate significant differences between density groups of the same hybrid.

**Figure 3 animals-11-02292-f003:**
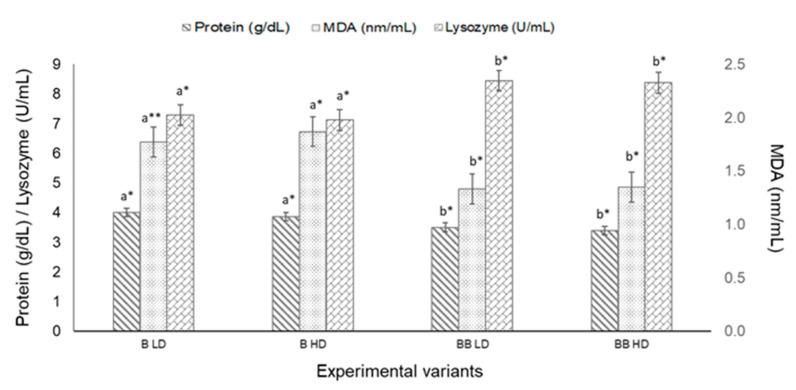
Serum protein, MDA, and lysozyme concentrations of bester (B) and bester × beluga (BB) sturgeon hybrids, cultured in different initial stocking densities LD (low density—2 kg/m^2^) and HD (high density—4 kg/m^2^). Data are presented as mean ± SEM (*n* = 3 tanks per treatment). Different letters indicate significant differences between hybrids from the same density group (*p* < 0.05, two-way ANOVA). Different symbols */** indicate significant differences between density groups of the same hybrid.

**Table 1 animals-11-02292-t001:** Growth performance and feed utilization of sturgeon hybrids (B and BB) reared under different stocking densities.

Parameter	Bester (B)		Bester × Beluga (BB)	
	LD	HD	LD	HD
Initial density (kg/m^2^)	2	4	2	4
Final density (kg/m^2^)	3.1	5.9	3.5	6.7
IW (g)	181.47 ± 1.06 ^a^*	182.48 ± 0.94 ^a^*	183.87 ± 1.68 ^a^*	180.10 ± 0.31 ^a^*
IL (cm)	27.62 ± 0.58 ^a^*	27.54 ± 0.43 ^a^*	26.21 ± 0.54 ^a^*	26.00 ± 0.49 ^a^*
FW (g)	289.73 ± 1.93 ^a^**	269.42 ± 6.06 ^a^*	322.63 ± 6.66 ^b^**	308.60 ± 4.47 ^b^*
FL (cm)	34.60 ± 0.36 ^a^*	33.10 ± 0.40 ^a^*	32.76 ± 0.68 ^a^*	31.93 ± 1.25 ^a^*
WG (%)	59.66 ± 1.99 ^a^**	47.62 ± 2.16 ^a^*	75.56 ± 1.78 ^b^*	71.30 ± 3.11 ^b^*
CvBWi (%)	5.72	4.87	9.2	6.9
CvBWf (%)	18.81	16.39	29.12	24.03
SGR (%/day)	1.23 ± 0.03 ^a^**	1.02 ± 0.04 ^a^*	1.48 ± 0.04 ^b^*	1.41 ± 0.07 ^b^*
FCR	0.85 ± 0.08 ^a^**	0.97 ± 0.07 ^a^*	0.74 ± 0.09 ^b^**	0.85 ± 0.11 ^b^*
PER	2.17 ± 0.01^a^**	1.91 ± 0.01 ^a^*	2.88 ± 0.01 ^b^**	2.18 ± 0.01 ^b^*
K	0.87 ± 0.07 ^a^*	0.85 ± 0.03 ^a^*	0.92 ± 0.18 ^b^*	0.90 ± 0.07 ^b^*
VSI (%)	6.72 ± 0.19 ^a^**	6.03 ± 0.22 ^a^*	6.91 ± 0.29 ^b^*	6.89 ± 0.31 ^b^*
HSI (%)	2.21 ± 0.19 ^a^**	1.82 ± 0.15 ^a^*	2.13 ± 0.17 ^b^*	1.98 ± 0.14 ^b^*

Data are presented as mean ± SEM (*n* = 3 tanks per treatment). Different letters in the same row indicate significant differences between hybrids from the same density group (*p* < 0.05; two-way ANOVA). */** Different symbols in the same row indicate significant differences between stocking density groups of the same hybrid.

**Table 2 animals-11-02292-t002:** Hematological parameters of two hybrid sturgeons reared under different stocking densities for 6 weeks.

Parameter	Bester (B)		Bester × Beluga (BB)	
	LD	HD	LD	HD
RBC (10^6^/mL)	0.43 ± 0.10 ^b^**	0.58 ± 0.13 ^a^*	0.44 ± 0.06 ^b^**	0.62 ± 0.14 ^b^*
PCV (%)	29.16 ± 3.12 ^a^*	29.10 ± 2.09 ^a^*	22.52 ± 3.90 ^b^**	20.87 ± 3.30 ^b^*
Hb (g/dL)	6.57 ± 0.57 ^a^**	6.17 ± 0.60 ^a^*	5.96 ± 0.98 ^b^*	5.86 ± 0.51 ^b^*
MCV (fl)	707.65 ± 136.77 ^a^**	520.59 ± 94.39 ^a^*	524.56 ± 114.93 ^b^**	351.07 ± 85.35 ^b^*
MCH (pg)	159.39 ± 29.31 ^a^**	110.29 ± 21.17 ^a^*	139.14 ± 29.59 ^b^**	98.12 ± 16.00 ^b^*
MCHC (g/dL)	22.68 ± 2.43 ^a^**	21.18 ± 0.79 ^a^*	26.63 ± 2.56 ^b^**	28.55 ± 3.27 ^b^*

Data are presented as mean ± SEM (n = 3 tanks per treatment). Different letters in the same row indicate significant differences between hybrids from the same density group (*p* < 0.05, two-way ANOVA). */** Different symbols in the same row indicate significant differences between density groups of the same hybrid.

**Table 3 animals-11-02292-t003:** Growth parameters of BB hybrid reared under various stocking densities.

Parameter	SD1	SD2	SD3	SD4	Equation	R^2^	*p*
Initial density (kg/m^2^)	5	8	10	12	-	-	*-*
IW (g)	327.50 ± 7.12	329.55 ± 5.26	326.85 ± 8.24	323.86 ± 4.56	-	-	*-*
CvBWi (%)	9.55	8.14	9.98	10.89	-	-	*-*
IL (cm)	33.25 ± 0.87	33.38 ± 0.98	33.28 ± 1.23	33.167 ± 1.45	-	-	*-*
Final density (kg/m^2^)	9	13	16	20	-	-	-
FW (g)	608.35 ± 10.17	546.50 ± 12.56	526.35 ± 13.89	523.21 ± 12.23	y = 2.312x^2^ − 49.070x + 800.930	0.99	0.008
CvBWf (%)	22.76	24.77	26.72	31.72	y = 1.217x + 15.841	0.89	0.048
FL (cm)	39.13 ± 1.17	39.04 ± 0.89	38.44 ± 0.95	39.23 ± 1.35	y = 0.025x^2^ − 0.432x + 40.805	0.33	0.770
WG (%)	85.75 ± 3.13	65.83 ± 2.77	61.03 ± 4.22	61.55 ± 5.83	y = 0.612x^2^ − 13.427x + 134	0.99	0.037
SGR (%/day)	1.63 ± 0.11	1.33 ± 0.15	1.25 ± 0.43	1.26 ± 0.34	y = 0.009x^2^ − 0.205x + 2.383	0.99	0.046
FCR	0.91 ± 0.04	0.94 ± 0.02	1.11 ± 0.04	1.15 ± 0.05	y = 0.037x + 0.69	0.88	0.038
PER	2.03 ± 0.08	1.96 ± 0.07	1.67 ± 0.10	1.61 ± 0.07	y = −0.066x + 2.402	0.89	0.050
K	0.92 ± 0.03	0.91 ± 0.01	0.91 ± 0.03	0.91 ± 0.01	y = 0.0004x^2^ − 0.008x + 0.951	0.74	0.098
VSI (%)	7.19 ± 0.43	6.96 ± 0.29	6.12 ± 0.04	4.67 ± 0.07	y = −0.071x^2^ + 0.858x + 4.687	0.99	0.008
HSI (%)	2.92 ± 0.14	2.91 ± 0.68	2.61 ± 0.95	2.23 ± 0.98	y = −0.021x^2^ + 0.270x + 2.118	0.84	0.040

Data are presented as mean ± SEM (*n* = 2 tanks per treatment).

**Table 4 animals-11-02292-t004:** Effects of different stocking densities on the values of hematological indices and serum biochemical parameters of the BB hybrids.

Parameter	SD1	SD2	SD3	SD4	Equation	R^2^	*p*
RBC (10^6^/mL)	0.52 ± 0.10	0.69 ± 0.11	0.82 ± 0.29	0.86 ± 0.39	y = 0.050x + 0.280	0.97	0.015
PVC (%)	21.95 ± 6.63	21.86 ± 1.74	23.04 ± 1.86	24.90 ± 4.47	y = 0.108x^2^ − 1.425x + 26.343	0.99	0.033
Hb (g/dL)	6.08 ± 0.68	6.28 ± 2.70	6.24 ± 0.53	6.69 ± 0.40	y = 0.013x^2^ − 0.144x + 6.4984	0.86	0.366
MCV (fl)	430.95 ± 126.70	325.27 ± 49.33	301.75 ± 78.18	323.37 ± 112.75	y = 5.060x^2^ − 101.510x + 812.27	0.99	0.021
MCH (pg)	116.74 ± 17.66	74.31 ± 39.48	81.75 ± 21.71	87.54 ± 26.73	y = 2.110x^2^ − 39.602x + 260.87	0.83	0.201
MCHC (g/dL)	29.34 ± 9.90	23.71 ± 11.78	27.15 ± 2.35	27.53 ± 4.86	y = 0.293x^2^ − 5.109x + 47.235	0.67	0.781
MDA (g/dL)	1.03 ± 0.19	0.88 ± 0.11	0.97 ± 0.07	1.37± 0.13	y = 0.026x^2^ − 0.4002x + 2.378	0.99	0.009
Protein (g/dL)	4.57 ± 0.23	4.11 ± 0.08	3.87 ± 0.07	4.07 ± 0.08	y = 0.025x^2^ − 0.509x + 6.498	0.95	0.023
Glucose (mg/dL)	88.14 ± 0.79	88.98 ± 0.51	89.64 ± 0.88	90.81 ± 1.06	y = 0.370x + 86.147	0.96	0.089
Cortisol (ng/mL)	42.48 ± 0.21	42.51 ± 0.27	43.82 ± 0.24	43.91 ± 0.43	y = 0.020x^2^ − 0.106x + 42.425	0.81	0.431
Lysozyme (U/mL)	10.50 ± 0.65	9.23 ± 0.22	8.81 ± 0.21	8.42 ± 0.25	y = −0.295x + 11.824	0.95	0.032

Data are presented as mean ± SEM (*n* = 2 tanks per treatment).

## Data Availability

All the data are available from the first author, and can be delivered if required.

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
