# Peer review of "Effects of Stocking Density on Growth Performance and Stress Responses of Bester and Bester ♀ × Beluga ♂ Juveniles in Recirculating Aquaculture Systems"

_animals, 2021, doi:10.3390/ani11082292_

Round 1
Reviewer 1 Report
The Authors of the manuscript entitled "Effects of Stocking Density on Growth Performance and Stress Responses of Bester and Bester ♀ x Beluga ♂ Juveniles in Recirculating Aquaculture Systems" have brought forward an interesting study revolving around the topic of sturgeon production. These Chondrosteans are commonly recognized as fish of high value, but are also quite difficult to maintain and breed, especially larger species which develop functional gonads as late as after 10+ years. This is why the search for either technological or methodical improvements, which could significantly improve the breeding or rearing of these living fossils remains crucial for the growth of this industry.
This manuscript was written well, predominantly, and is surely worthy to be published, but I have some objections regarding the M&M section (information missing) and the quality of presentation of the Results (graphs require corrections and visual improvement). I have listed my comments below, paragraph by paragraph.
Simple summary: This part was written very well.
Abstract: This paragraph is clear and concise.
Keywords: I have no objections here. Eventually, one more keyword could be added, namely "serum parameters", but it is not that important.
Introduction: This is a very nicely written paragraph, all of the aspects of the study have been introduced in a logical order, but without dragging it out too much. I have only found minor mistakes, all of which I have outlined below.
Line 47: Add a comma after "diversification".
Line 50: Change to plural "sizes".
Line 52: Change "operation" to "later operate".
Line 53: Add "it is" after "as".
Line 56: Change "proofing" to "to enhance".
Lines 57-58: Please restructure the phrase "in special in long period holding species like" to "especially in species maintained for extended periods such as".
Line 64: Delete "the" before "females".
Line 70: Change "in which both represent" to ", both of which represent.
Lines 72-87: I believe that the scientific name of the sterlet should appear somewhere in this paragraph, like it was given for the beluga.
Line 94: Change "effects process" to "cascade of signals,".
Line 102: Add a comma after "method".
Lines 107-108: Please add the scientific names of the two species, the Atlantic and Amur sturgeons, in parentheses.
Materials and Methods: I did not find any serious methodological errors in this part, the study was apparently well planned and the measurements/analyses were not too complicated, so there was not much room for misconception. There is some important information missing from the experimental description, though, which definitely needs to be added. Please follow the remarks below.
Line 136: I believe it should be "9 aquaculture breeders" (4 females + 5 males).
Line 139: Add "they" before "reached".
Line 141: Add "they" before "were".
Lines 143 and 146: Were the BB hybrids sorted for growth, and bigger specimens were taken for T2, while smaller ones for T1? Please specify that.
Line 146: What is this "g/exemplar" supposed to indicate? I believe that it should rather be "g/specimen". Furthermore, are those 8 tanks the same 1 m3 tanks as in T1? Please, specify that and also indicate the dimensions of these tanks.
Line 150: Was the feeding 2% BW daily, or 2% BW per each of the three meals? I believe 2% daily is definitely the obvious answer, but still this is a bit unclear here.
Line 152: Delete "the" after "if".
Line 153: How many fish were measured weekly from each tank/group? I believe this is an important piece of information since sturgeons are very likely to stress out when taken out of water, even for a short period of time, especially when held still (anaesthetized?) for length measurements.
Lines 156-167: Were these water quality measurements conducted separately for each tank? It appears to be the case, but I just wonder whether the automated monitoring system had probes for each separate tank, since all tanks were a part of a single RAS.
Lines 169-181: The calculation of HSI and VSI indices implies that the sturgeons had to be dissected. However, it is not specified anywhere in this paragraph if all experimental fish were euthanized, then measured for length and body weight, and then dissected for liver and visceral weight measurements, or if only some of the fish were killed off and dissected. Please indicate this clearly.
Line 193: Correct to "Zeiss".
Line 208: Capitalize the start of the new sentence "In brief ...".
Line 209: There is something missing here in the phrase "a volume of 0 .01%", likely a verb. Please clarify.
Line 211: Correct to "absorbance".
Line 221: Correct to "Duncan's".
Results: The presentation of obtained data is overally sufficient, clear and straightforward, but still could be much improved. Most of all I believe that the aesthetics of the Figures could be improved significantly by unifying them in terms of color coding for groups, style formatting etc. Other issues were outlined below.
Lines 227-234: First of all, there is no indication about mortality (even if there was none). Furthermore, it has not been indicated properly if there was a statistically significant difference in final BW between B and BB hybrids, neither in this paragraph or in Figure 1 (SD bars are not shown either).
Line 249: Correct to "were found".
Lines 254-258: I suggest to move this paragraph below Fig. 2, to improve the paper's formatting and to move the description of Fig. 3 after the introduction of Fig. 2.
Figure 2: This Figure contains only one set of significance signs "a/b" and "*/**", which is misleading because it is not clear whether they are attributed to cortisol or glucose measurements. Following the text, it appears that the markings for glucose are the ones which are missing. Anyway, please correct this issue. I would suggest to change this Figure into a bar graph only, alike Figure 3. Also, please correct the description of the Figure: "Different */* symbols ...".
Figure 3: Like above, please correct the description of the Figure: "Different */* symbols ...".
Line 307: Correct to "were found".
Line 308: Correct to "(p < 0.05)".
Figure 5: I have the same objections as for Figure 2 - only a single set of significance letters is given, all the while there are two parameters being displayed. Once again, I suggest to remake this figure into a bar graph for both parameters, like Figure 3 was done. Furthermore, the groups are coded as "SD", while in the description (and also in Figures 4 and 6) they are called "D". In other various places these coding fluctuates between "D" and "SD", therefore, please unify the group coding for T2 throughout the manuscript.
Figure 6: This time there are all three sets of letters given, but I still believe that using just bars would make this figure stylistically consistent with the other graphs (the line between points suggests some continuity between them, change over time etc.; meanwhile this is entirely not true).
Discussion: I have no major objections for this part, as every result has been well discussed and numerous references to other studies were given as well. I managed to find only minor errors, but nothing substancial.
Line 342: Correct to "oxygen".
Line 357: The scientific name of the Atlantic sturgeon should have already appeared in the Introduction.
Line 358: Correct the references, it is twice "55".
Line 363: Correct to "hybrids".
Line 366: Correct to "induce".
Line 379: Move comma behind "juveniles".
Line 397: Correct to "T1 and T2".
Line 415: Add the common species name "Chinese sturgeon".
Line 437: Change "from" to "in".
Line 462: Correct to "disturbing".
Line 467: Add the scientific names of the three species in parentheses.
Line 477: Correct to "HSI".
Line 479: Correct "leading to" to "are".
Conclusions: Clear and concise, no objections here.
Reviewer 2 Report
I have included remarks and comments to the manuscript in the attached file. The manuscript should be redesigned and rewritten in accordance with the comments, especially in terms of the analysis and presentation of the results obtained. In its present form, it is not suitable for publishing. I suggest resubmitting to Animals after general completing and redrafting the text.

Reviewer 3 Report
This manuscript describes the effects of stocking densities on the growth performance and physiological parameters in two hybrid species of the sturgeon. The experiments were well planned with 2-3 replicates, results were clear, and the conclusions were supported by the results except for a point described in the separated comment sheets. Overall, the manuscript contains valuable information for the sturgeon aquaculturists. Minor revision is needed.

Round 2
Reviewer 1 Report
The Authors of the manuscript entitled "Effects of Stocking Density on Growth Performance and Stress Responses of Bester and Bester ♀ x Beluga ♂ Juveniles in Recirculating Aquaculture Systems" have closely followed the suggestions of all reviewers and thoroughly revised their paper, from start to the very end. To the Authors' credit, all of my previous remarks have been addressed and I especially like the improvement shown in the Materials & Methods and Results sections. All of the missing information in M&Ms was added properly, while the Tables and Figures in the Results were reworked and now contain no visible mistakes. There are only minor issues that need to be corrected, as indicated below.
I must admit that during the first round review I did not notice that there even was a supplementary file attached to the manuscript (containing the technical scheme of the RAS), which definitely proves to be a great addition to the article. It is good that the Authors added the proper information about this Figure in the text, but it should also be repeated after the Conclusions - Animals obligates to include this mandatory section named "Supplementary materials" (see the template file in the Instruction for Authors).
Furthermore, in regard to this final, obligatory part of every manuscript, the paragraph in Lines 137-141 (ethics statement) should be moved from the main text (or may be copied) into the "Institutional Review Board Statement", at the end, below the "Supplementary materials" (again, see the template file). Additionally, a "Data Availability Statement" is missing here as well. Please, pay attention to the journal's obligatory formatting of this section.
Other than these little technical issues, which should be corrected in the final editorial draft, I believe that this paper is now fully acceptable for publishing. Good work!
Author Response
Dear Reviewer,
Please find below the responses to your suggestions.
Sincerely,
Mirela CREȚU
